



# Ice layer formation in the snowpack due to preferential water flow: case study at an alpine site

Louis Quéno[1], Charles Fierz[1], Alec van Herwijnen[1], Dylan Longridge[1], and Nander Wever[2]

[1]WSL Institute for Snow and Avalanche Research SLF, Davos, Switzerland
[2]Department of Atmospheric and Oceanic Sciences, University of Colorado Boulder, Boulder, CO, USA

**Correspondence:** Louis Quéno (queno@slf.ch)

**Abstract.** Ice layers may form in the snowpack due to preferential water flow, with impacts on the snowpack mechanichal, hydrological and thermodynamical properties. This case study at a high-altitude alpine site aims at monitoring their formation and evolution during winter 2017 thanks to the combined use of a comprehensive observation dataset at daily frequency and detailed snowpack modelling. In particular, daily SnowMicroPen penetration resistance profiles enabled to better iden-

tify ice layer temporal and spatial heterogeneity when associated with traditional snowpack profiles and measurements, while upward-looking ground penetrating radar measurements enabled to detect the water front and better describe the snowpack wetting when associated with lysimeter runoff measurements. One-dimensional snowpack simulations with the SNOWPACK model successfully represented the formation of some ice layers when using Richards equation and preferential flow domain parameterization, with a newly implemented ice reservoir. Detailed snowpack simulations with snow microstructure repre-

sentation, associated with high-resolution comprehensive observation dataset were shown relevant for studying and modelling such complex phenomena, despite limitations inherent to 1D modelling.

## 1   Introduction

The presence of ice layers in a snowpack may impact its mechanical, hydrological and thermodynamical properties. Monitoring the formation and evolution of ice layers is thus crucial in many research fields. Because of their low permeability (Albert and

Perron Jr., 2000), ice layers may increase the liquid water storage of the snowpack, which can substantially affect the snowpack runoff (Singh et al., 1999). On the contrary, near-surface ice layers in Greenland were shown to prevent access to deeper firn layers, thus reducing meltwater storage in the firn and enhancing ice sheet mass loss (Machguth et al., 2016). The stability of a mountainous snowpack may also be affected, with a possible increased faceting of the microstructure close to ice or crusts (Jamieson, 2006; Hammonds et al., 2015; Hammonds and Baker, 2016). Moreover, retrieval algorithms for water equivalent

of snow cover and snow depth from passive microwave emissions are sensitive to the presence of ice layers (Rees et al., 2010; Roy et al., 2016). Better knowledge about their formation could help the assimilation of such data in detailed snowpack models (Larue et al., 2018).

   Ice forms in the snowpack can have different origins (Fierz et al., 2009). They can either form at the surface, because of freezing rain (Quéno et al., 2018) or firnspiegel formation process due to radiative cooling (Ozeki and Akitaya, 1998), or they



can form within the snowpack, through percolation of rain or meltwater reaching subfreezing snow (Pfeffer and Humphrey, 1998). The present study focuses on the latter case. As opposed to matrix water flow, leading to a homogeneous progression of the wetting front, preferential water flow occurs through flow fingering (e.g. Schneebeli, 1995), transporting liquid water to deeper regions of the snowpack where the cold content is sufficient to refreeze it (e.g. Marsh, 2006). Preferential flow occurs in the snowpack due to its microstructural heterogeneity (density, grain size and shape) at layer transitions (Katsushima et al.,
2013), which may trigger flow fingering or form hydraulic barriers (i.e. capillary or permeability barriers) where water ponds and may subsequently refreeze. Hydraulic barriers may also divert water flow and lead to lateral flow along slopes (Eiriksson et al., 2013; Webb et al., 2018a). Knowledge about water percolation in snow, and particularly preferential flow, has recently made great progress in terms of process understanding and numerical simulation. Using dye tracer and liquid water content (LWC) measurements, Avanzi et al. (2016) observed preferential flow and water ponding at capillary barriers for various layer
transition characteristics and water input. X-ray microtomography was also used to observe wet-snow metamorphism under preferential flow (Avanzi et al., 2017). Magnetic Resonance Imaging observations of finger flow and lateral flow emphasized that even small differences in snow properties may form capillary barriers in dry snow (Katsushima et al., 2018). At larger scale, the effect of preferential flow on the snowpack runoff was assessed, through measurements of the heterogeneity of water discharge (Yamaguchi et al., 2018; Webb et al., 2018b), or under rain-on-snow conditions (Würzer et al., 2017; Juras et al.,
2017). The new insights from measurement campaigns enabled the development of multi-dimensional models accounting for preferential flow in the snowpack (Hirashima et al., 2014, 2017, 2019; Leroux and Pomeroy, 2017, 2019). They also enabled progress in the representation of water transport in one-dimensional models. Richards equation was implemented in detailed snowpack models like SNOWPACK (Wever et al., 2014) and Crocus (D'Amboise et al., 2017), as an improvement over the more simplistic bucket parameterization, enabling a more realistic representation of water transport in snow with
respect to snow microstructure. Based on recent studies relating preferential flow to snow properties (Katsushima et al., 2013; Hirashima et al., 2014; Yamaguchi et al., 2012), with analogies to preferential flow in soils (DiCarlo, 2007, 2013), Wever et al. (2016) developed an original one-dimensional parameterization of preferential flow in SNOWPACK through a dual-domain implementation separating matrix flow and preferential flow, both solved with Richards equation.

       The present study builds on the work of Wever et al. (2016), who investigated ice layer formation at an alpine site thanks to
a new modelling scheme parameterizing preferential water flow in the SNOWPACK model, with comparisons to manual snow profiles every two weeks over 16 winter seasons. We push forward this investigation through a case study including several novelties. First, a comprehensive observation dataset was gathered at the same research site, in order to better determine the evolution of the snowpack and identify the formation of deep ice layers in natural conditions at an high altitude alpine site. The originality of this dataset comes from the opportunity to monitor ice formation in natural alpine conditions during a
whole winter season at daily resolution, even though the present study does not include detailed observations of preferential water flow paths in cold laboratory. This dataset is used for a detailed assessment of the preferential flow representation in SNOWPACK, bringing complementary insights to Wever et al. (2016) and Würzer et al. (2017). An effort is also made to improve the simulation of heterogeneous deep ice formation in the SNOWPACK model.



The paper is organized as follows. Section 2 describes the study site, the observation dataset, the snowpack simulation
configuration and new methods. Section 3 details the results with insights on water percolation and ice layer formation from
both the observation dataset and the simulations. Their benefits and limitations are discussed in Sect. 4.

## 2   Data and methods

### 2.1   Observation dataset

The site of study is the Weissfluhjoch (WFJ) measurement site, a research field dedicated to snowpack investigations, located at
an elevation of 2536 m.a.s.l. above Davos in the Eastern Swiss Alps (Marty and Meister, 2012). For comparison to simulations,
we use a comprehensive observation dataset collected during winter 2017. Figure 1 indicates the location of the measurements
in the research field. Traditional snowpack profiles were performed during the entire season every week (or every two weeks at
the beginning and the end of the season), along three corridors. These profiles were carried out following the recommendations
of Fierz et al. (2009) and gather observations of the grain shape, grain size, layer thickness, hand hardness index and wetness
through visual and manual assessment, as well as measurements of snow depth (HS), water equivalent of snow cover (SWE),
snow density, snow temperature, snow hardness (Swiss ramsonde), and liquid water content with a Denoth device (Denoth,
1989). Continuous melt-freeze crusts (MFcr) and ice layers (IFil) were identified in the snow profile, while ice lenses were
most often noted additionally. There was no measurement of ice layer density, as such measurements in natural conditions
remain very challenging (e.g. Watts et al., 2016). Snowpack runoff measurements were provided by a 5 m$^2$ lysimeter at 10 min
temporal resolution. Finally, thermistores at a fixed vertical interval of 20 cm provided half-hourly snow temperature profiles
of the snowpack (Fierz, 2011).

An upward-looking ground penetrating radar (upGPR) was also installed on the site (Schmid et al., 2014). The dual frequency
GPR from IDS (Ingegneria Dei Sistemi, Italy) conducted measurements every 30 minutes and with two different frequencies,
600 MHz and 1.6 GHz. Every measurement contained 1800 traces with 1024 samples. In order to get rid of system ringing, a
hoisting device was installed by Schmid et al. (2014) to move the GPR antennas vertically during a measurement cycle. When
transmitting electromagnetic waves into snow, discontinuities result in reflections, refractions and diffractions. The amount
of energy reflected at the discontinuity is proportional to the relative change across the discontinuity (Reynolds, 2011). The
percolated water changes the internal properties of the snow. The boundary between wet and dry snow (called water front
hereafter) appears as a distinct reflector and can then be determined by semi-automatic picking, similarly to the algorithm
developed by Schmid et al. (2014) for snow surface picking. The picked two-way travel times of the water front are multiplied
by the wave propagation velocity of 0.23 m ns$^{-1}$, which is a typical value for snow (Sand and Bruland, 1998; Schmid et al.,
2014). During winter 2017, the water front could be derived until a technical malfunction on 9 April prevents further analysis.
Due to the 45° angle of beam spread of the upGPR, the footprint in the snowpack where the water front is derived can measure
a homogeneous state but not local flow fingers.

In addition, daily measurements of the penetration resistance were performed with a SnowMicroPen (SMP; Schneebeli et al.,
1999), in the context of the RHOSSA field campaign (Calonne et al., 2019), launched in winter 2016. The SMP has a tip surface



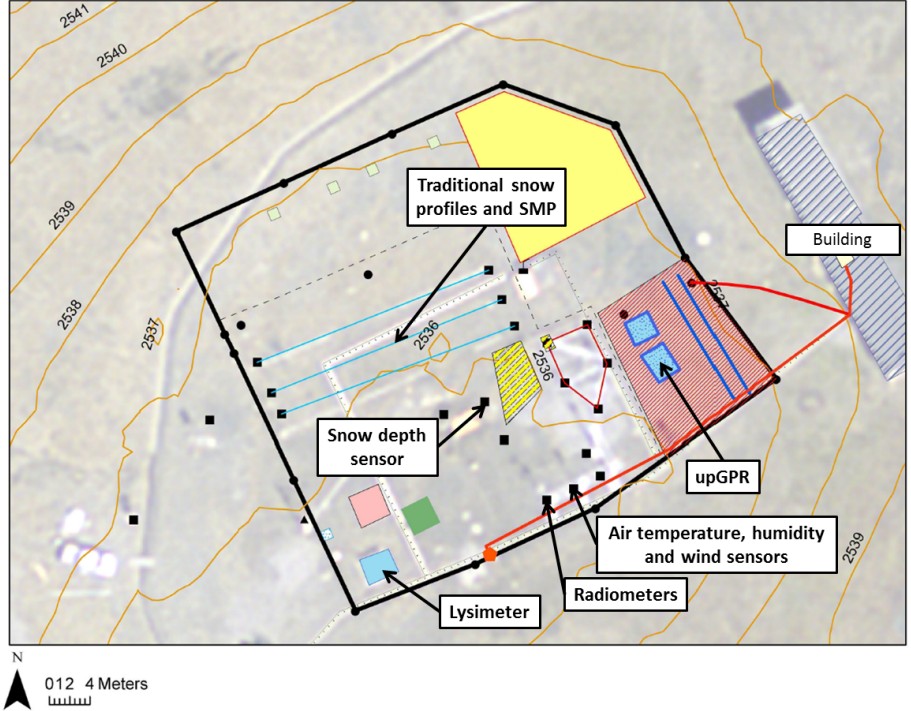

**Figure 1.** Overview of the WFJ research field, with contour lines in orange and labels indicating the location of the measurements used in this study.

of 19.6 $mm^2$. Every day during the winter season, five to seven SMP measurements were performed along the corridors, with a 15 cm spacing perpendicular to the direction of the corridor, providing indications about the local spatial heterogeneity of potential ice forms. A limited number of gaps in the daily measurements have to be noted, in particular from 25 March to 27

March and 7 and 8 April in the wetting period.

For longer-term observations of ice layers, we gathered traditional snowpack profiles performed every two weeks during the winter seasons 1999/2000 to 2015/2016.

## 2.2 Snowpack simulations

### 2.2.1 Simulation configuration

SNOWPACK simulations were performed for the WFJ study site in winter 2017. They were initialized with a traditional profile recorded on 3 January 2017, to provide a realistic base of the snowpack with a snow depth of 47 cm. The initialization date was chosen early enough to assess the model ability to simulate the microstructure evolution as well as water percolation. The simulations were driven by optimal in situ meteorological measurements (Fig. 1) of air temperature and humidity (ventilated sensors), near-surface wind, solar and longwave irradiance (WSL Institute for Snow and Avalanche Research SLF, 2015; Wever



et al., 2015). The snowfall input is driven by measured snow depth increments (Wever et al., 2015), enabling direct comparisons to measurements and outcomes of Wever et al. (2016).

Three water transport schemes implemented in SNOWPACK were evaluated. First, the bucket approach (BA) is a common method used in snowpack models (e.g. Bartelt and Lehning, 2002; Vionnet et al., 2012), assuming that water is transported to the next downward layer when the liquid water content exceeds the water holding capacity of a given layer (depending

on the ice volumetric content of snow; Coléou and Lesaffre, 1998). Second, the Richards equation (RE) was implemented in SNOWPACK by Wever et al. (2014) to account for capillary effects. These effects are modelled taking into account the water retention curve (van Genuchten, 1980) and the hydraulic conductivity of snow (Mualem, 1976; van Genuchten, 1980; Calonne et al., 2012). Third, a dual-domain approach parameterizing preferential flow and using Richards equation (RE/PF) was recently developed. Water exchanges between the matrix domain and the preferential flow domain are determined according to the water

entry pressure head in the matrix layers and the saturation in the preferential flow domain: this implementation is described in details in Wever et al. (2016). The two tuning parameters of this scheme were chosen here accordingly to Wever et al. (2016): the threshold in saturation of the preferential flow domain $\Theta_{th} = 0.08$ and the parameter related to the number of flow paths per square meter $N = 0$. In particular, $N = 0$ implies no refreezing of the preferential flow water (Wever et al., 2016). Similarly to Wever et al. (2016), SNOWPACK simulations are carried out at high vertical resolution, with a layer merging threshold of 0.25

cm and new snow layer initialization of 0.5 cm. High resolution is necessary to permit the formation of very thin high density layers.

### 2.2.2 Implementation of ice reservoir

A new parameterization of ice layer formation due to preferential flow was implemented as a complement to the RE/PF scheme. It is summarized in Fig. 2. In the RE/PF scheme, when the saturation in the preferential flow domain exceeds the threshold

$\Theta_{th}$, water flows back to the matrix domain depending on the freezing capacity of the matrix domain; if the threshold is still exceeded then, saturations in both domains are equalized (Wever et al., 2016). Thus, the part of water that freezes in the matrix domain is spread in the whole layer, while ice lenses may only form locally at the base of the flow fingers.

To overcome this issue, we developed an ice reservoir parameterization. The water normally transferred from the preferential flow domain to the matrix domain that freezes instantly is stored in an ice reservoir (step 4 in Fig. 2), instead of being added

to the ice volumetric content of the matrix. The ice reservoir is representative of the volumetric content of ice lenses (i.e. spatially heterogeneous ice) in a given layer. The transferred water that does not freeze goes in the matrix domain , i.e. is spread homogeneously (step 5 in Fig. 2).

Furthermore, the saturation threshold in the PF domain (Wever et al., 2016) was chosen as a simple solution to the unability of Richards equation to model the saturation overshoot present in the tip of flow fingers (DiCarlo, 2007). This simple

parameterization can lead to inconsistencies due to the vertical discretization of the simulated snowpack. After water has been transferred to the matrix at the layer corresponding to the finger tip (i.e. where the saturation threshold was exceeded), the highest saturation is then reached more likely at the layer above, where no water transfer occurred. Because of that, the water transfer from PF domain to matrix domain may spread over too many layers, instead of being concentrated in the lowest





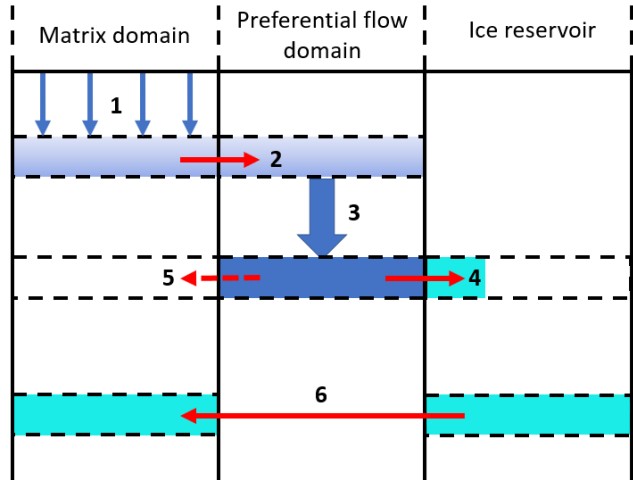

**Figure 2.** Scheme of the ice reservoir parameterization in SNOWPACK. Blue represents liquid water, cyan represents ice and red arrows represent water or ice transfers to another domain. Steps 1 to 6 are described in the text.

layer (i.e. the tip of the flow finger). To overcome this issue, the ice reservoir was cumulated in the lowest layer. When the
ice volumetric content of the cumulated ice reservoir added to the ice volumetric content and water volumetric content of the
associated matrix layer exceeds the corresponding ice density threshold of $700 \, \mathrm{kg \, m^{-3}}$ in SNOWPACK, there is enough ice
to consider it as horizontally homogeneous: the ice content of the cumulated ice reservoir is then transferred to the associated
matrix layer (step 6 in Fig. 2). Simulations with the ice reservoir parameterization are called RE/PF/IceR hereafter.

## 3 Results

### 3.1 Insights from the observation dataset

#### 3.1.1 Overview of the winter season

At WFJ, winter 2016/2017 started with a shallow snowpack of approximately $30 \, \mathrm{cm}$ at the beginning of November, followed
by a extended period of calm weather, forming a base layer made of depth hoar crystals (Richter et al., 2019). This layer was
covered by new snow at the end of December, and several small snow storms lead to a maximum snow depth of $205 \, \mathrm{cm}$ on 10
March, i.e. slightly lower than the average maximum snow depth. The snowpack had entirely melted on 14 June. Overall, this
winter was characterized by lower snow depth than long-term averages in the region.

Figure 3 represents the evolution of grain types within the snowpack as observed in traditional snow profiles. Two layers
of particular interest are tracked during the whole winter. Layer 1 corresponds to surface hoar formed at the end of January.
This layer is continuously identified as buried surface hoar (as primary or secondary grain) until the end of March, with a grain
size substantially larger than the layer above (consisting of rounded grains or faceted crystals), thus constituting a capillary





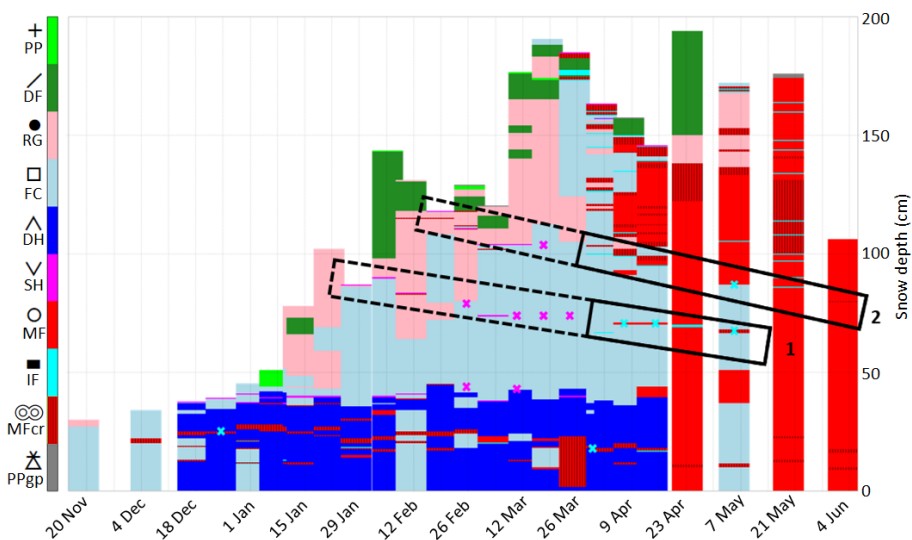

**Figure 3.** Visual observations of grain shapes at WFJ during winter 2017. Colours, symbols and codes are defined following the grain shape classification of Fierz et al. (2009), fuchsia crosses represent surface hoar as secondary grain shape, cyan crosses represent ice lenses. Rectangles highlight layers 1 and 2, with dashed lines before ice formation and solid lines afterwards.

barrier with a classical fine-over-coarse structure (e.g. Avanzi et al., 2016). Ice is observed above this barrier from 28 March, either as a homogeneous layer or as ice lenses mixed with melt forms (red in Fig. 3). Thicknesses between 0.5 cm and 1 cm were reported. The presence of nearby slopes (NNW of the snow profile corridors in Fig. 1) may suggest a water input through lateral flow along the capillary barrier. Simulations (not taking into account lateral flow) may provide complementary

insights to determine whether vertical preferential flow was sufficient to form an ice layer. Layer 2 corresponds to surface hoar appearing in mid-February, and forming a capillary barrier once buried. An ice layer is observed at that level in most profiles after 28 March.

### 3.1.2   Water percolation and snowpack runoff

Figure 4 represents lysimeter measurements of the snowpack runoff together with the height of the water front estimated

from the upGPR measurements, from 1 March to 15 April, i.e. the transition period from dry to isothermal snowpack. Shown underneath are the snow temperature measurements in the lowest meter of the snowpack, at fixed intervals of 20 cm. The height of water front could only be derived until 8 April due to technical issues. Before 30 March (first dashed line), no snowpack runoff was observed, the water front was always higher than 1 m, and temperatures in the lowest first meter of the snowpack were all below $0^\circ$C. Between 30 March and 9 April (second dashed line), the water front remained high (mostly higher than

1 m), a small amount of snowpack runoff was observed (less than $3 \, \mathrm{kg \, m^{-2} \, day^{-1}}$). Snow temperatures gradually increased to reach $0^\circ$C by the end of the period in the lowest meter of the snowpack. From 9 April, all temperatures in the lowest meter reach $0^\circ$C and snowpack runoff increases. Although after 9 April no more water front estimates were available from the upGPR



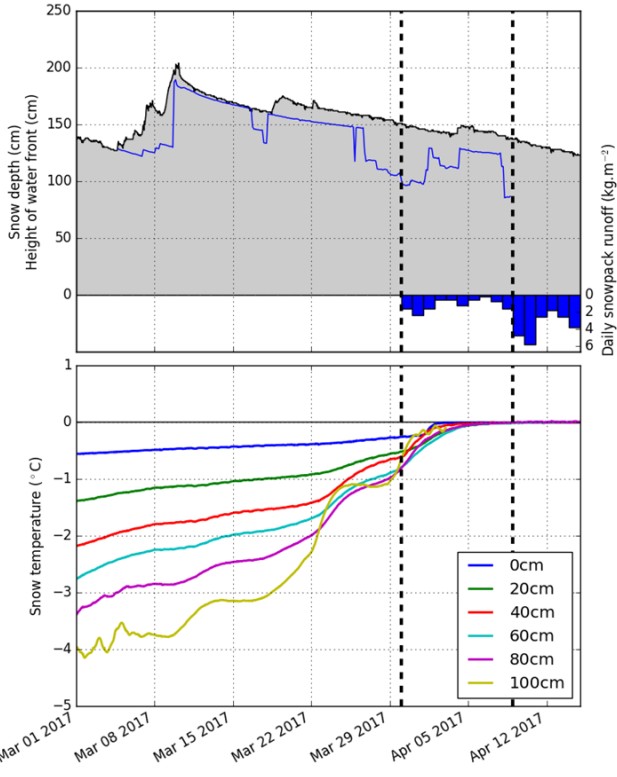

**Figure 4.** Top: evolution of the snow depth (black line), height of the water front (blue line) and daily snowpack runoff (blue bars), from 1 March 2017 to 15 April 2017 at WFJ. Bottom: measured snow temperatures at different heights above the ground, same period, same location. Dashed lines indicate 30 March and 9 April.

data, it reached the lowest value (85 cm) on 8 April. The snowpack runoff is yet low compared to the more significant runoff starting in mid-May, as shown in Fig. 5: the first 20 $\mathrm{kg\,m^{-2}}$ of total snowpack runoff are reached on 10 April, while the first day with a daily snowpack runoff higher than 10 $\mathrm{kg\,m^{-2}\,day^{-1}}$ is 13 May.

These measurements give insights in the timing of water percolation in the snowpack. Before 9 April, the bottom of the snowpack was cold and dry while the water front was still mostly above 100 cm. The low snowpack runoff values were thus likely due to preferential flow paths reaching the ground. After 9 April, the lowest meter of the snowpack reaches an isothermal state at 0°C: snowpack runoff increases markedly and is likely due to matrix flow reaching the ground.

### 3.1.3 Capillary barriers and ice layers

To study the formation of ice through preferential flow in subfreezing snow, we focus on the lowest meter of the snowpack. In this part, the manual snow profiles enable to identify three main capillary barriers (Fig. 3): the two layers of buried surface hoar where ice forms at the end of March (defined as layers 1 and 2 previously) and the top of the depth hoar base layer, where


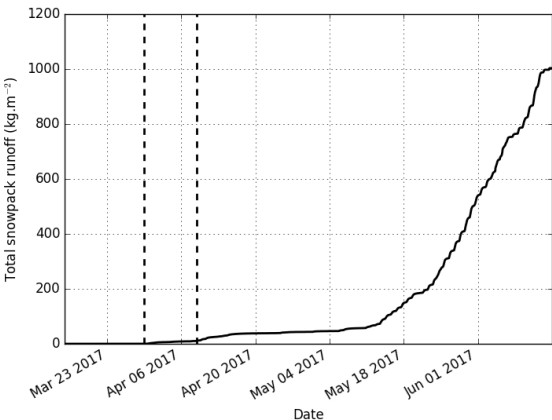

**Figure 5.** Total snowpack runoff from 15 March 2017 to 15 June 2017 at WFJ. Dashed lines indicate 30 March and 9 April.

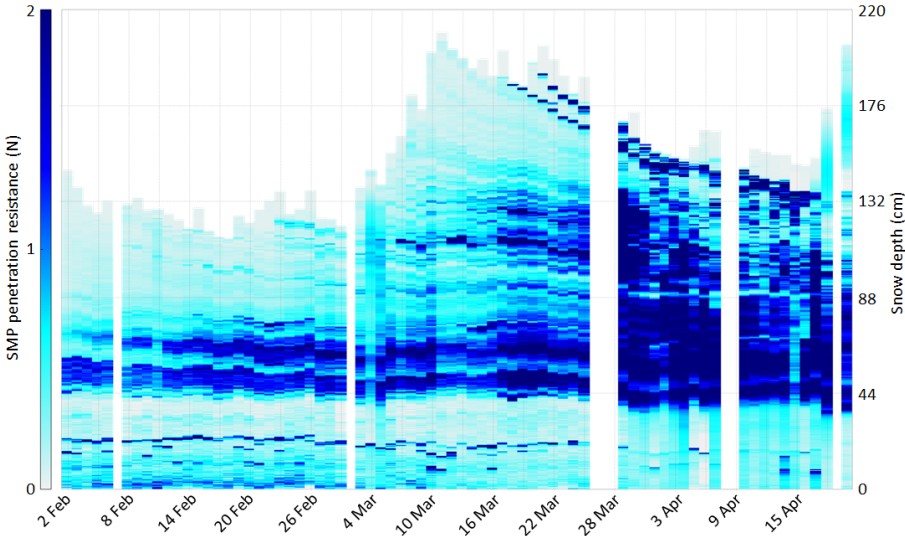

**Figure 6.** Daily SMP measurements of penetration resistance (mm averaged values) at WFJ from 1 February to 19 April, with one representative profile per day.

higher water content is observed in April and May. Layers 1 and 2 are marked by grain size heterogeneity with the overlying layers: on 28 February, 1 mm over 2.5 mm for layer 1, 0.5 mm over 2 mm for layer 2.

Daily SMP measurements enable to more clearly identify the temporal and spatial variability of ice formation. Figure 6 represents the evolution of penetration resistance from 1 February to 19 April, with a scale from 0 N to 2 N to highlight variations in dry snow. The deep MFcr is visible in the middle of a low resistance depth hoar layer, at approximately 20 cm. In March, the buried surface hoar of layer 1 is marked by a lower penetration resistance than the surrounding faceted crystals.

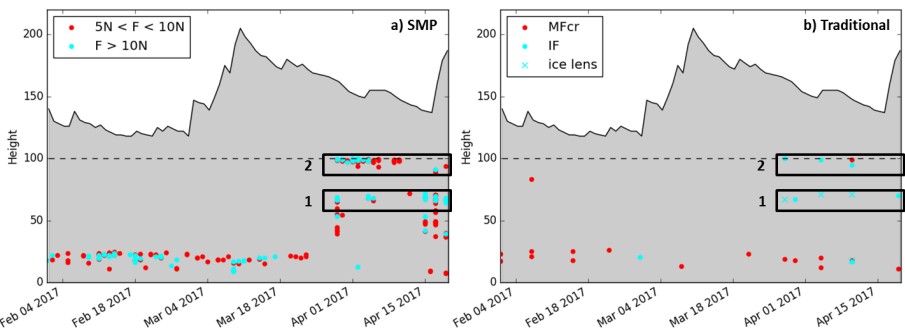

**Figure 7.** a) Height of SMP penetration resistances between 5 N and 10 N (red) and higher than 10 N (cyan) below 1 m, from 1 February to 19 April at WFJ. b) Visual observations of melt-freeze crusts (MFcr), ice layers (IFil) and ice lenses, same period and location. Daily manual measurements of snow depth in solid black line. Rectangles highlight layers 1 and 2.

Layer 2 exhibits less heterogeneity with surrounding layers. Overall, the penetration resistance increases substantially from the end of March on. Figure 7a represents penetration resistances higher than thresholds of 5 N and 10 N, below 100 cm, including all daily SMP measurements. These values were chosen after comparison of matching traditional and SMP profiles, to better highlight crusts and ice forms. Except the deep MFcr mostly visible until the end of March, two high resistance layers can be identified from 29 March: they match the visual identification of ice layers 1 and 2 (Fig. 7b). They can be tracked until

mid-April, but are not present on all SMP profiles.

  The penetration resistance measured at these two layers was tracked in the SMP profiles. To identify these layers, all SMP profiles were superimposed on the traditional profile observed at the closest date, compared to each other for a given day and the next day. The value of SMP penetration resistance was then visually picked. Figure 8 shows an example of this procedure for two SMP profiles of 4 April 2017.

The evolution of the penetration resistance of the two tracked layers is plotted in Fig. 9 from 14 March to 19 April. For layer 1, the penetration resistance remains very low (less than 1 N) until 24 March. It corresponds to the observed layer of buried surface hoar. On 29 March, all seven SMP measurements show penetration resistance higher than 10 N indicating the continuous presence of ice. Afterwards, values alternate between low resistance (less than 5 N) and very high resistance (more than 10 N), as visible in Fig. 8 on 4 April. This is consistent with the visual observations reporting a layer in which pure

ice and melt forms are both observed. These observations suggest that water ponding at the capillary barrier did not freeze everywhere on the study plot where we performed these measurements. For layer 2, very low penetration resistances are also measured until 24 March, corresponding to the observed buried surface hoar. After 29 March, values increase (mostly higher than 5 N, often more than 10 N), indicating formation of ice. After 9 April, no more high resistances are measured, rather low resistances corresponding to a wet layer of melt forms. The evolution of penetration resistances for layer 2 show more temporal

consistency than layer 1, suggesting that the ice layer disappears totally after 9 April.





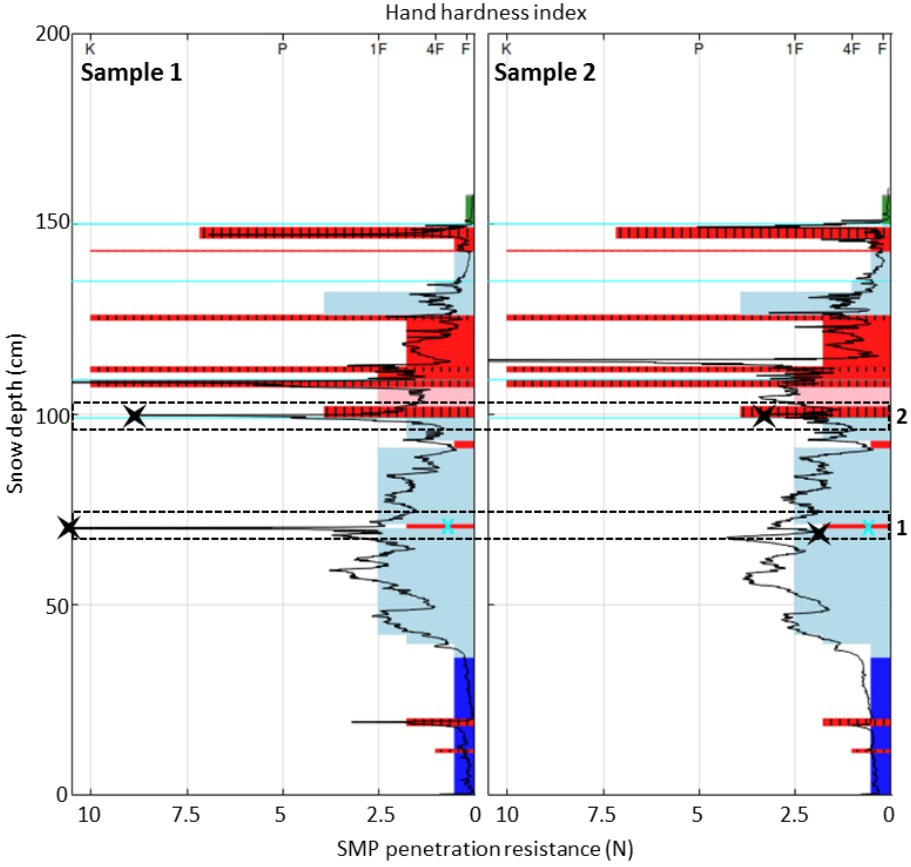

**Figure 8.** Two samples of SMP profiles of 4 April 2017 (black line), superimposed on the traditional snow profile of the same date. Colours referring to the grain shape and hand hardness index are defined accordingly to the classification of Fierz et al. (2009). Dashed rectangles highlight layers 1 and 2. Black crosses indicate the penetration resistance picked for each layer.

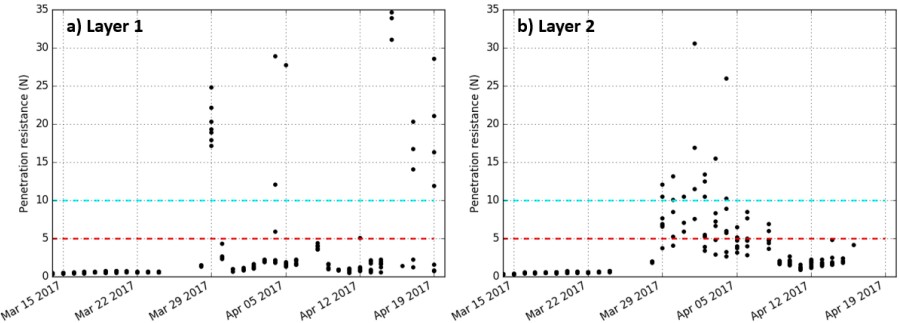

**Figure 9.** Evolution of the penetration resistance of a) layer 1 and b) layer 2, manually tracked in the SMP profiles, from 14 March to 19 April. Thresholds of 5 N and 10 N are indicated in red and cyan respectively, accordingly to Fig. 7.

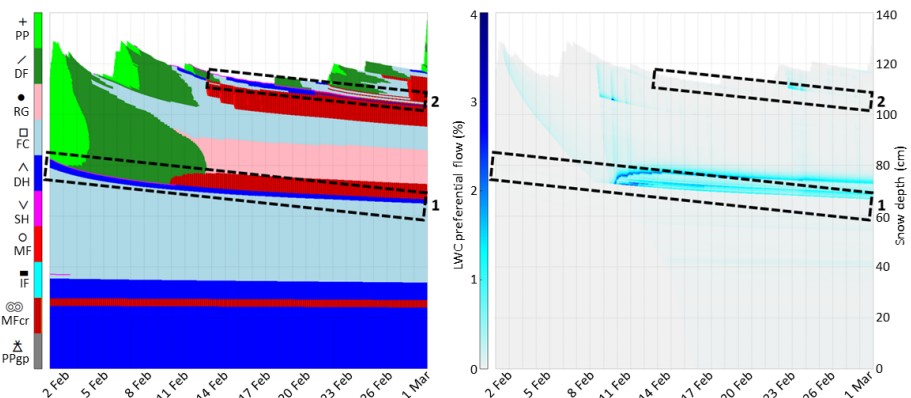

**Figure 10.** SNOWPACK simulation RE/PF, February 2017 at WFJ: a) grain shape, according to the classification of Fierz et al. (2009); b) liquid water content in the PF domain. Rectangles highlight layers 1 and 2.

## 3.2 Assessment of snowpack simulations

### 3.2.1 With different water transport schemes

Simulations of winter 2017 were first performed with the three water transport schemes existing in SNOWPACK (BA, RE and RE/PF). For the RE/PF simulation, Fig. 10 shows the grain shape and liquid water content in the PF domain for the month of

February. Buried surface hoar of layer 1 (represented in fuchsia, at approximately 80 cm) is well simulated at the beginning of February. The capillary barrier of layer 2 is also well initiated in mid-February on the surface. However, a thick melt-freeze crust forms at layer 1 on 10 February (represented in hatched red, Fig. 10a). It is associated with some melting close to the surface leading to preferential water flow refreezing at capillary barrier of layer 1 (Fig. 10b). The water transferred for refreezing in the matrix domain is spread homogeneously which forms a crust with a density of approximately 320 $\mathrm{kg\,m^{-3}}$.

The transfer spreads vertically due to the issues mentioned in Sect. 2.2.2. This thick melt-freeze crust was not observed in the manual profiles nor the SMP measurements. Other melt-freeze crusts form close to the surface from mid-February. They were observed (Fig. 3) but were thinner than the simulated ones. A little surface melting is simulated on 1 February and leads to preferential flow (Fig. 10). Contrary to later simulated melting in mid-February, it was not observed: the measured snow surface temperature remained slightly under melting point. This simulation error is likely due to excessive surface turbulent

fluxes input. New simulations were run without this melt water input, with no effect on the later snowpack structure due to the limited melting amount.

Figure 11 shows the grain shape and the liquid water content in the matrix domain from 15 March to 15 April, i.e. the period of transition from dry to ripe snowpack when ice layers formed (Sect. 3.1). No ice layer forms, except at the snowpack basis, which is probably due to a boundary effect at the interface between snow and soil. The thick melt-freeze crust is still

present at layer 1, thus no ice layer forms (Fig. 11a). However, a higher water retention is simulated (Fig. 11b). Due to the excessive formation of melt-freeze crusts, the simulated snow microstructure at layer 2 does not reproduce the observed snow





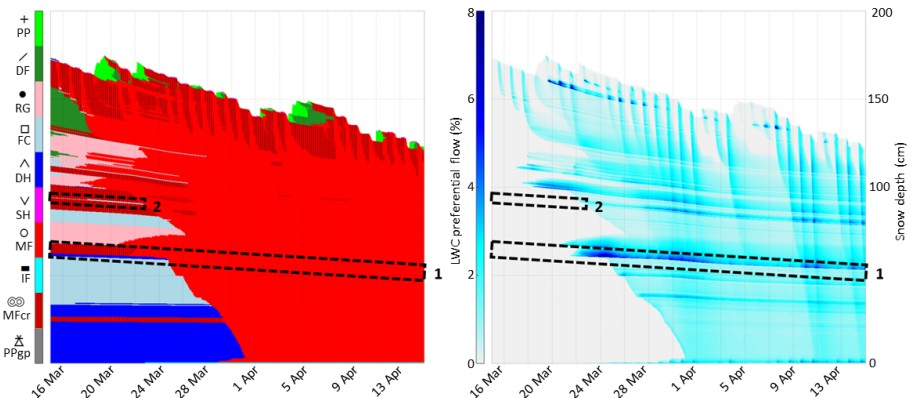

**Figure 11.** SNOWPACK simulation RE/PF, from 15 March to 15 April 2017 at WFJ: a) grain shape, according to the classification of Fierz et al. (2009); b) liquid water content in the matrix domain. Rectangles highlight layers 1 and 2.

microstructure and the capillary barrier, thus no ice layer forms. Matrix flow reaches the ground and the snowpack is entirely isothermal on 31 March (Fig. 11b), i.e. 9 days before the observations (Sect. 3.1.2).

A sensitivity study on the parameters of the dual-domain approach ($\Theta_{th}$ and $N$) was performed, but could not resolve

the issues about ice formation highlighted here. Simulations were also analyzed in terms of snowpack runoff, confirming earlier findings (Wever et al., 2016; Würzer et al., 2017). The BA scheme underestimates the snowpack runoff, the RE scheme overestimates it, and the addition of preferential flow increases this overestimation (not shown). The onset of snowpack runoff is delayed compared to observations for BA and RE, because preferential flow is not simulated, but RE/PF simulations show the onset of snowpack runoff too early.

### 3.2.2 With ice reservoir parameterization

Simulations were also performed with the ice reservoir parameterization (RE/PF/IceR) to assess its ability to improve the simulation of ice layer formation compared to the previous simulations. Figure 12 shows the grain shape and liquid water content in the PF domain for the month of February. Contrary to the RE/PF simulation, no melt-freeze crust forms at layer 1 (Fig. 12a): the water leaving the PF domain and refreezing is in too low quantity to be considered as representative of the mean

state of the snowpack in this layer, it is thus stored in the ice reservoir. The fine-over-coarse grain transition forming a capillary barrier is preserved. Note that liquid water content in the PF domain (Fig. 12b) is almost not modified compared to the RE/PF simulation (Fig. 10b). Liquid water transport is similar, and in particular the vertical spreading of water ponding, but ice in the reservoir is concentrated at the capillary barrier. At the end of February, less melt-freeze crusts are formed than in the RE/PF simulation, even though the ones surrounding layer 2 are also thicker than observed.

Figure 13 shows the grain shape and the liquid water content in the matrix domain from 15 March to 15 April. A basal ice layer forms similarly to the RE/PF simulation. Contrary to the RE/PF simulation, the capillary barrier of layer 1 is still present (Fig. 13a). Melt forms appear at the layer transition from 22 March, and an ice layer forms in the matrix domain on 24 March,



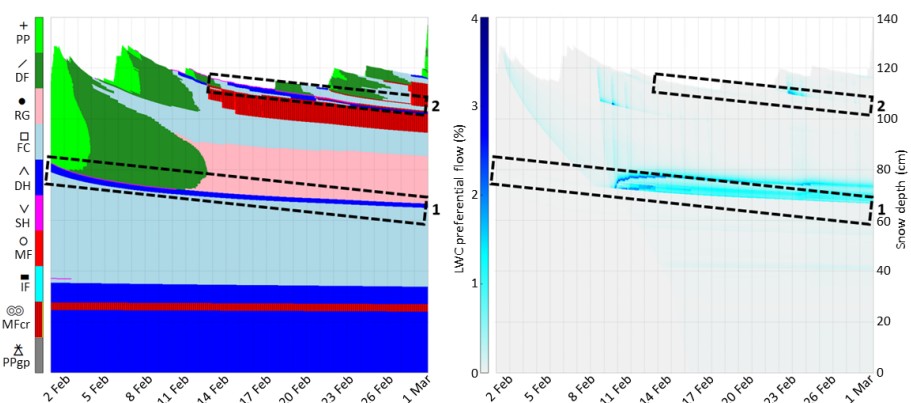

**Figure 12.** SNOWPACK simulation RE/PF/IceR, February 2017 at WFJ: a) grain shape, according to the classification of Fierz et al. (2009); b) liquid water content in the PF domain. Rectangles highlight layers 1 and 2.

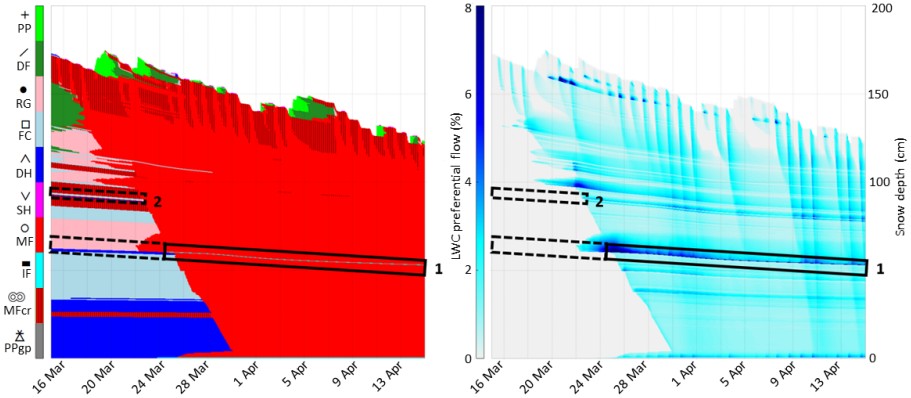

**Figure 13.** SNOWPACK simulation RE/PF/IceR, from 15 March to 15 April 2017 at WFJ: a) grain shape, according to the classification of Fierz et al. (2009); b) liquid water content in the matrix domain. Rectangles highlight layers 1 and 2, with dashed lines before ice formation in the matrix domain and solid lines afterwards.

i.e. 4 to 5 days earlier than observed (Sect. 3.1.1 and Sect. 3.1.3). At this date, ice is transferred from the ice reservoir to the matrix domain (Fig. 14). The ice layer formed is 43 mm thick, with a dry density of 821 $kg\,m^{-3}$ and a significant volumetric

liquid water content of $\theta_{matrix} = 7.8\%$ and $\theta_{PF} = 1.7\%$ (on 24 March 13 UTC). Similarly to the RE/PF simulation, no ice forms at layer 2 due to the presence of melt-freeze crusts. However, less melt-freeze crusts are simulated in the snowpack, which is more in accordance with the observations. The matrix water flow reaches the ground on 30 March (Fig. 13b), i.e. 10 days before the observations (Sect. 3.1.2). The ice reservoir does not modify the snowpack runoff compared to RE/PF simulations (not shown).



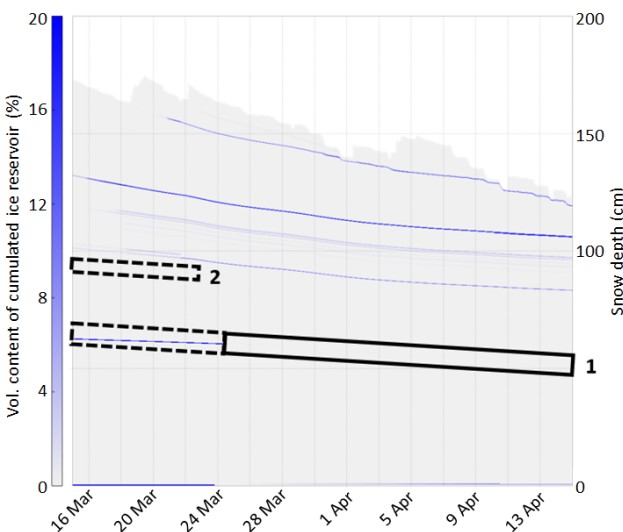

**Figure 14.** SNOWPACK simulation RE/PF/IceR, from 15 March to 15 April 2017 at WFJ: volumetric ice content in the cumulated ice reservoir. Rectangles highlight layers 1 and 2, with dashed lines before ice formation in the matrix domain and solid lines afterwards.

### 3.2.3 Simulations over several winter seasons

To assess the impact of the ice reservoir parameterization on ice formation in the SNOWPACK model, simulations at WFJ are performed over 17 winters from 1999/2000 to 2015/2016, using the RE/PF and the RE/PF/IceR configurations. Only traditional snow profiles are available for evaluation, similarly to Wever et al. (2016). Ice layers at the snow-soil interface are not taken into account because of changing snowpack base in the observations over the winters (wooden board, gravel) and possible boundary effects in the simulations. Only simulated ice layers are verified against observations, to calculate hits (number of simulated ice layers matching an observation) and false alarms (number of simulated ice layers that do not match any observation). A height difference of 20 cm is used for matching of simulations and observations, similarly to Wever et al. (2016). We assume the formation date of an observed ice layer is comprised between the last snowpack profile without observed ice layer and the first profile where it is indicated. If the simulation date is more than one month away from the observed formation time interval, or if the height difference is more than 20 cm, the event is considered as a false alarm. Results of this multi-year evaluation are summarized in Table 1.

Overall, the addition of the ice reservoir parameterization to the RE/PF scheme enables to form more ice layers, with a higher number of hits (15 against 6, with a one month tolerance) and a lower number of false alarms (1 against 6). The simulated ice formation date is in average 22 days earlier than the observation interval for the RE/PF scheme and 4 days earlier for the RE/PF/IceR scheme. The RE/PF/IceR configuration also mitigates the number of unobserved early melt-freeze crusts compared to the RE/PF configuration (not shown), as already highlighted for season 2016/2017 (Fig. 10 and Fig. 12). However, a very high number of ground ice layers were simulated (17 for the RE/PF/IceR scheme against 8 for the RE/PF scheme). This number is probably excessive, and due to possible snow-soil boundary effects.





**Table 1.** Hits (HI) and False Alarms (FA) of simulated ice layers for RE/PF and RE/PF/IceR simulations at WFJ, for 17 winter seasons, from 1999/2000 to 2015/2016. HI (height and date): simulated ice layers that match an observed one formed at less than 20 cm of height difference and in the same time interval. HI (height only): simulated ice layers that match an observed one formed at less than 20 cm of height difference and less than one month away from the observed time interval. FA: simulated but not observed ice layers (or more than one month away from the simulated formation).

|            | HI (height and date) | HI (height only) | FA |
|------------|:--------------------:|:----------------:|:--:|
| RE/PF      | 3                    | 3                | 6  |
| RE/PF/IceR | 5                    | 10               | 1  |

## 4  Discussion

This case study of ice layer formation at Weissfluhjoch enables to assess both a comprehensive observation dataset and detailed 1D snowpack simulations for monitoring a complex process. We discuss hereafter the relevance of these methods and results.

First, the combined use of traditional snow profiles with measurements of higher temporal resolution like the SMP provides a suitable observation framework to study the transition period from dry to isothermal snowpack when ice formations appear due to preferential flow. Snowpack runoff measurements associated with snow temperature sensors and upGPR-derived water front

gave insights about the homogeneous wetting of the snowpack and the period when the bottom of the snowpack was primarily affected by preferential flow. SMP profiles of penetration resistance showed clear signals of ice presence, when compared to visual observations, with values higher than 10 N (Fig. 7) while penetration resistances in dry snow were mostly lower than 2 N (Fig. 6) and melt-freeze crusts were characterized by intermediate penetration resistances (usually between 5 N and 10 N, Fig. 7). Identification of ice layers with several SMP profiles regularly spaced also offers a more quantitative estimate of ice

heterogeneity than a subjective visual observation. However, the temporal and spatial variabilities may be complex to distinguish, as shown for layer 1 (Fig. 9). The visual layer tracking of SMP profiles is also a source of uncertainties. Hagenmuller and Pilloix (2016) developed a method matching several hardness profiles to synthesize them into one representative profile. This method was not considered relevant for the present study which focuses on local heterogeneity of ice layers. Moreover, when ice layers are too thick, the SMP cannot go through them as happens often in spring. For winter 2017 at WFJ, no complete

SMP profile could be performed after 19 April. Finally, the difficulty to identify the exact date of ice formation with biweekly traditional snowpack profiles (Sect. 3.2.3) highlights the added value of the more comprehensive observation dataset used for winter 2016/2017. Overall, this comprehensive high-resolution dataset (also including detailed measurements of density and specific surface area of the snow; Calonne et al., 2019) provides valuable information for a thorough validation of today's and future snow cover models.

One-dimensional SNOWPACK simulations provide complementary insights to the observation dataset, despite the spatial heterogeneity of ice layer formation due to preferential flow. The addition of an ice reservoir enables to parameterize the local formation of ice at capillary barriers: it may thus be considered as representative of the volumetric content of ice lenses at a





given layer. These local specificities are not taken into account in the matrix domain, which is the mean state of the snowpack, until they become homogeneously spread. Several limitations may be noted. The matrix flow modeled by Richards equation

305 occurs too early and leads to an excessive snowpack runoff, which is even more enhanced by preferential flow. This may explain the too early formation of ice layer 1: the matrix water front reaches this level on 24 March in simulations, while it is observed higher than 1 m on 24 March and the surrounding layers are still dry on 28 March in the observations when the first ice lenses are observed. In addition, the performance of the RE/PF water transport scheme strongly depends on a good representation of the snow microstructure by SNOWPACK, and particularly the grain radius and snow density. For instance, no ice nor water

310 ponding are simulated at layer 2 during winter 2017 because the observed capillary barrier structure (rounded grains and faceted crystals above surface hoar) is not adequately represented (unobserved melt-freeze crusts above layer 2). Finally, the implementation of the ice reservoir is meant to improve the representation of ice formation within the 1D framework of the RE/PF dual-domain approach, but it does not mitigate the limitations of the preferential flow parameterization. In particular, the vertical spreading of water flowing back from preferential to matrix domain is not solved: its effect on ice formation is only

315 mitigated with the cumulated ice reservoir. Advances in preferential water flow modelling in the snowpack have recently been developed by Leroux and Pomeroy (2017, 2019) to tackle the capillary hysteresis effect and capillary pressure overshoot. They could be considered to improve the representation of preferential flow in the SNOWPACK model, through a more accurate determination of capillary pressure at the tip of the preferential flow path, with effects on the water transfer from preferential flow to matrix domain.

320 Despite the limitations inherent to 1D simulation of preferential flow, the dual-domain approach combined to the ice reservoir parameterization in SNOWPACK provides relevant information concerning deep ice layer formation. The ice reservoir limits the formation of unobserved early melt-freeze crusts and, overall, enables to simulate more observed ice layers. For the case study of winter 2017, it gives complementary insights on the formation of ice layer 1: according to the simulations, the vertical preferential flow was sufficient to form the ice layer, even though a possible contribution of lateral flow cannot be totally

325 excluded.

## 5 Conclusions

We presented here a case study of ice layer formation in the snowpack due to preferential water flow at Weissfluhjoch, a high-altitude alpine site. Monitoring deep ice layers is of particular relevance for many applications, but is challenging in natural snow conditions. This research proposed an approach based on the combined use of a novel comprehensive observation dataset

330 at high temporal resolution and detailed snowpack modelling, during winter 2017.

Weekly traditional snow profiles, snowpack runoff and temperature measurements as well as upGPR-derived water front height enabled to better monitor the dry-to-wet transition period between mid-March and mid-April 2017. In particular, the first days of measured snowpack runoff could be attributed to preferential water flow, and the exact date of first isothermal snowpack with matrix water flow reaching the ground could be identified. Daily penetration resistances measured by SMP

335 gave more accurate insights on ice layers, in complement to the traditional visual observations. Through comparisons with the

visual observations, penetration resistance thresholds of 5 N and 10 N in SMP profiles could be defined for the identification of melt-freeze crusts and ice layers, respectively. Ice formation could be monitored at higher temporal resolution, and the use of several profiles per day gave more quantitative information on ice spatial heterogeneity. The daily succession of profiles also enabled to track the two main capillary barriers where ice formed, providing additional information on the evolution of the

layers.

One-dimensional SNOWPACK simulations, including a parameterization of preferential flow, showed an overall good representation of the snowpack structure, but a too early matrix wetting associated with an excessive snowpack runoff. The observed ice layers were not simulated due to the early formation of thick melt-freeze crusts, explained by limitations of the preferential flow scheme. We developed an ice reservoir parameterization to mitigate these limitations, with freezing water transferred

from the preferential flow domain to a reservoir. The ice was included in the matrix domain when the layer could be considered to be continuous. This parameterization improved the simulation of winter 2017 with a limited number of unobserved early melt-freeze crusts and the formation of one ice layer. However, the early transition to a wet snowpack was not improved, as the water transport was not modified. The ice reservoir scheme also showed improvements for the simulation of ice layers over past seasons.

These simulations highlighted the relevance of detailed snowpack models for the modelling of complex phenomena like ice layers formed by preferential water flow, since an accurate representation of the snow microstructure is necessary. Recent advances in preferential flow observations and modelling could contribute to strengthen water transport representation. This project also underlined the importance of comprehensive observation datasets for the validation of complex snow models. Collecting high-resolution data over more winter seasons should improve the understanding of deep ice layer formation,

particularly concerning their density, their impermeability and their evolution in the late melting season.

*Code and data availability.*   The dataset used in the paper will be available on the EnviDat database (doi will be provided). The ice reservoir parameterization for SNOWPACK will be available on models.slf.ch.

*Author contributions.*   CF and AvH designed the study, carried out the field measurements and were responsible for the maintenance of the equipment. LQ was responsible for the modelling strategy, analyzed the measurements and simulations and wrote the manuscript. DL

processed the upGPR data to estimate the location of the water front. CF, AvH and NW helped to analyze measurements and simulations. All authors contributed to the manuscript.

*Competing interests.*   The authors declare that the research was conducted in the absence of any commercial or financial relationships that could be construed as a potential conflict of interest.



*Acknowledgements.* We thank all the people from WSL Institute for Snow and Avalanche Research SLF involved in the measurements at
Weissfluhjoch.



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
