# Peer review of "Deep ice layer formation in an alpine snowpack: monitoring and modelling"

_The Cryosphere, 2020_

## Referee Comment (RC1) · Anonymous Referee #1 · 24 Mar 2020

This study provided very important data by field observation for water infiltration and ice layer formation using traditional snowpack observation, SnowMicroPen, and upper GPR. Considering water transport, thermal process, preferential flow, and the freezing process is important to study ice layer formation in the snow physics. Due to the difficulty of laboratory experiments for ice layer formation, the water transport model lacked validation data considering freezing. They are useful data to collaborate with laboratory experiments and 2D or 3D models. I would like to know whether authors have a plan to open these data. The implementation of ice reservoir parameterization is also interesting. And I impressed that the ice layer was reproduced well using a one-dimensional model. In my opinion, this paper is suitable to accept for The Cryosphere. I suggest several comments to make this paper more informative. But it is not necessary

requirement for acceptance. Please refer to comments to make this paper be better.

minor comment

Line 123-144 Section 2.2.2 is a new parameterization and main improvement part of the model. So a more detailed expression is necessary. I guess the ice in the matrix layer block (or transport with very low permeability) for water transport. But it is unclear whether the ice in the ice reservoir also affects hydraulic conductivity or not. Please add the description about the influence of ice in ice reservoir on water transport simulation.

Line 164-166 Is the SNOWPACK simulation reproduced this water front observed by upGPR, runoff and temperature profiles well?

Line 186-188 Fig.6 showed the mean SMP resistance. Can you explain the heterogeneity of SMP resistance other than layer 1 and 2 briefly?

Line 196-199 Confirmation of melt-freeze crust on buried surface hoar seems valuable observation result. Considering the theory, a capillary barrier forms on surface hoar due to low suction of surface hoar and froze later. Is it confirmed past study or observations? If so, please add the reference.

Fig 10-13 Please add a and b in the figure.

Line 247. I felt that the Fig. 12b is strange. In Fig. 10b, water ponds at 70-80cm height (on the freezing layer) on 11 February. On the other hand, despite the frozen layer was not shown in Fig 12a, water ponds at the same place. I guess that ice exists in the ice reservoir which was not shown in Fig. 12, and it affects this ponding. If my guess is correct, ice content in the ice reservoir had better be shown in some way. Also, can you describe the influence of the ice content in the ice reservoir on water percolation? In 2.2.2, formation of ice in ice reservoir is described. But influence of it on water transport should be also explained.

Line 325 (4 discussion): This study is attractive for researchers of water transport mechanisms using laboratory experiments and 2D or 3D models. Also, collaboration

with them enhances the value of this study in terms of cover some assumptions and make physical evidence. I expect the observation data will be opened. Also, during conducting this study, if the author came up with the idea that could be done by a laboratory experiment or a 2D or 3D model, suggesting ideas in discussion will be a good information.

---

## Referee Comment (RC2) · Francesco Avanzi (Referee) · 4 Apr 2020

This study combines a valuable dataset made by snow-pit, SnowMicroPen, upGPR, lysimeter, and weather-snow measurements with improvements to the liquid-water-transport scheme of SNOWPACK to study the important, but still poorly understood topic of deep ice-layer formation due to preferential flow. Results are promising, in that they show increased realism of the dual-domain scheme of SNOWPACK in reproducing ice-layer formation. Another interesting aspect of the paper is the overview of insights from the observational dataset and in particular Section 3.1.3.

My assessment is that the paper is interesting for readers of TC and certainly moves our understanding of ice layers forward. Yet, I do have several comments that may

be useful to enhance the clarity and analyses of the manuscript (see below), so I recommend the Editor accept the manuscript after minor revisions.

GENERAL COMMENTS

1. The manuscript sometimes reads like a case study (Title, Abstract), and sometimes as an incremental contribution to Wever et al. 2016. I do believe that there are several novel points in paper, some of which are discussed at lines 49ff, while some others emerge in Section 2.2.2. Overall, however, these novel points remain somewhat implicit for general readers. I encourage authors to elaborate on framing both in the Introduction and in the Discussion to better highlight these points of novelty.

2. Relatedly, novelty could also be better streamlined by explicitly identifying a few research questions to be reported at the end of the Introduction. These research questions should be broad enough to be interesting for the general public and could dramatically increase the impact of the paper.

3. In Section 3.2.3, authors chose to restrict results in Table 1 to only simulated ice layers, without considering instances when observed ice layers were not simulated. Did I understand correctly? If so, I would suggest authors to also include these 'missed' ice layers in Table 1 to gain further insights.

SPECIFIC COMMENTS

- I would recommend authors avoid wording like 'case study' in the Title and elsewhere in the manuscript.

- In the abstract, I would report that the new parametrization allowed authors to significantly reduce overestimations of melt-freeze crusts (see Section 3.2). More generally, to me the abstract seems a little imbalanced toward explaining data and methods rather than findings and insights.

- Line 49: Between the previous paragraph (lines 23-48) and this one about your study (lines 49ff), I would add one more paragraph to specifically discuss previous attempts

of modeling ice layers and why more research on this topic is needed. What are the knowledge gaps that you are trying to fill with this impressive field campaign and your new parametrization of ice layers? This paragraph should help formulate research questions and therefore better make the case of this paper, especially since the discussion about snow models between lines 40 and 48 is general and not really focused on ice layers.

- Line 68: how are measurements along these three corridors managed? Do you use one corridor every other week, or simply use the same corridor every other week as long as you reach the end of it?

- Line 74: maybe cite Wever et al. 2014 when you introduce the snow lysimeter since Wever et al. 2014 also described the specific design of this instrument.

- Line 79: get rid of -> remove

- Line 100: why not initializing the simulation on a day with no snow on the ground?

- Line 106: did you still allow for mixed rain-snow or all-rain events? If yes, using which phase-partitioning method? I assume these are rare at WFJ, but still worth discussing given that the paper focuses on liquid-water-transport schemes

- Line 119: are carried out –> were carried out

- Line 125: could you add more details about how water flowing back to the matrix domain depends on the freezing capacity of the matrix domain? As a general note about this Section, I am missing some discussion on whether the ice reservoir affects hydraulic properties of the snow layer. This is important given that ice lenses often represent a hydraulic barrier.

- Line 136-137: It is a little unclear why the the highest saturation should be reached more likely at the layer above, where no water transfer occurred.

- Line 178-179: In Katushima et al. 2013 and then Avanzi et al. 2016, it is showed that

[Figure]

isothermal conditions may coexist with preferential flow. Given that snow pits showed vertical structural heterogeneity after April 9 (see Figure 3) and Hirashima et al. 2019 have showed that matrix flow is coupled with structurally homogeneous conditions, I would assume that matrix flow started well after April 9 at this site.

- Figure 6: maybe add the same black boxes as in Fig. 3 to denote location of ice layers? Also, the color scale is a little unclear to me: Figure 7 shows that resistance is locally well above 2N. Are all values above 2 N reported with the same color as 2 N?

- Line 198: how does this manual picking selection works and what is its uncertainty?

- Line 205: may this also suggest that the ice layer was subjected to periodical melt and freeze?

- Line 275: Could authors elaborate on reasons why the simulated ice formation date is in average 22 days earlier than the observation interval for the RE/PF scheme and 4 days earlier 275 for the RE/PF/IceR scheme?

---

## Author Comment (AC1) · 25 Jun 2020

**Answer to Referee #1**

We thank the referee for their insightful comments. Please find our detailed response to the issues raised by the reviewer below. Referee comments are in bold while our answers appear in blue. Changes in the manuscript appear in red (lines correspond to the marked-up manuscript). Please also refer to the marked-up manuscript.

**This study provided very important data by field observation for water infiltration and ice layer formation using traditional snowpack observation, SnowMicroPen, and upper GPR. Considering water transport, thermal process, preferential flow, and the freezing process is important to study ice layer formation in the snow physics. Due to the difficulty of laboratory experiments for ice layer formation, the water transport model lacked validation data considering freezing. They are useful data to collaborate with laboratory experiments and 2D or 3D models. I would like to know whether authors have a plan to open these data. The implementation of ice reservoir parameterization is also interesting. And I impressed that the ice layer was reproduced well using a onedimensional model. In my opinion, this paper is suitable to accept for The Cryosphere. I suggest several comments to make this paper more informative. But it is not necessary requirement for acceptance. Please refer to comments to make this paper be better.**

**minor comment**

**Line 123-144 Section 2.2.2 is a new parameterization and main improvement part of the model. So a more detailed expression is necessary. I guess the ice in the matrix layer block (or transport with very low permeability) for water transport. But it is unclear whether the ice in the ice reservoir also affects hydraulic conductivity or not. Please add the description about the influence of ice in ice reservoir on water transport simulation.**

The influence of ice in the matrix on water transport has not been changed from the original parameterization of Wever et al. (2016). On the other hand, the ice kept in the ice reservoir has no effect on the hydraulic properties of the snow layer. It is indeed a local reservoir, mimicking a heterogeneous layer formed of snow and ice lenses. Considering its effect on water transport may be out of scope for our 1D simulations, given our limited knowledge and data about local ice lenses hydraulic effect. This point has been clarified in the revised manuscript.

Please also note the implementation of the ice reservoir in the SNOWPACK source code will be made publicly available on models.slf.ch.

*--- CHANGES IN THE MANUSCRIPT (l. 154-176) ---*

*To better reproduce the formation of continuous ice layers from discontinuous and growing ice lenses, we developed an ice reservoir parameterization. The water normally transferred from the preferential flow domain to the matrix domain that freezes instantly is stored in an ice reservoir (step 4 in Fig. 2), instead of being added to the ice volumetric content of the matrix. The ice reservoir is representative of the volumetric content of ice lenses (i.e. spatially discontinuous ice) in a given layer. The transferred water that does not freeze goes in the matrix domain, i.e. is spread homogeneously (step 5 in Fig. 2).*

*Furthermore, the saturation threshold in the PF domain (Wever et al., 2016) was chosen as a simple solution to the unability of Richards equation to model the saturation overshoot present in the tip of flow fingers (DiCarlo, 2007). This simple parameterization can lead to inconsistencies due to the vertical discretization of the simulated snowpack. After water has been transferred to the matrix at the layer corresponding to the finger tip (i.e. where the saturation threshold was exceeded), the highest saturation is then reached more likely at the layer above, where no water transfer occurred, because water percolation from this layer to the finger tip layer only occurs at the next time step. Because of that, the water transfer from PF domain to matrix domain may spread over too*

*many layers, instead of being concentrated in the lowest layer (i.e. the tip of the flow finger). To overcome this issue, the ice reservoir was cumulated in the lowest layer. When the ice volumetric content of the cumulated ice*

*reservoir added to the ice volumetric content and water volumetric content of the associated matrix layer exceeds the corresponding ice density threshold of 700 kgm$^{-3}$ in SNOWPACK, there is enough ice to consider it as horizontally homogeneous: the ice content of the cumulated ice reservoir is then transferred to the associated matrix layer (step 6 in Fig. 2). As long as it is kept in the ice reservoir, the forming ice has no effect on the water transport in the matrix domain that still follows the RE/PF scheme (Wever et al., 2014, 2016). Furthermore, we neglect any impact the ice reservoir, which is interpreted as ice lenses, may have on hydraulic properties (e.g. local hydraulic barrier effect). Simulations with the ice reservoir parameterization are called RE/PF/IceR hereafter.*

*The implementation of the ice reservoir parameterization in the SNOWPACK source code is publicly available (see Code and data availability Section).*

**Line 164-166 Is the SNOWPACK simulation reproduced this water front observed by upGPR, runoff and temperature profiles well?**

As visible on Fig. 11b for the RE/PF configuration, the matrix water flow reaches deeper layers of the snowpack than the observed water front. For example, the water front is observed at around 100 cm in the beginning of April, while the simulated matrix flow reaches the ground at that time. The matrix flow for RE/PF/IceR simulations (Fig. 13b) is very similar to the RE/PF simulations. A reference to the water front observations has been added in that part of the text in the revised manuscript.

(Note that the y axis title of Fig. 11b and Fig. 13b was wrong: it is indeed the LWC in the *matrix* domain for the 15 Mar - 15 Apr period, as correctly formulated in the text and in the legend. It has been corrected.)

*--- CHANGES IN THE MANUSCRIPT (l. 269-271) ---*

*Matrix flow reaches the ground and the snowpack is entirely isothermal on 31 March (Fig. 11b), i.e. 9 days before the observations (Sect. 3.1.2). On 31 March, the water front was actually observed at around 100 cm (Fig. 4), hence a too early simulated water front progression.*

*--- CHANGES IN THE MANUSCRIPT (l. 297-298) ---*

*The matrix water flow reaches the ground on 30 March (Fig. 13b), i.e. 10 days before the observations (Sect. 3.1.2). The water front progression occurs too early, similarly to the RE/PF simulations.*

**Line 186-188 Fig.6 showed the mean SMP resistance. Can you explain the heterogeneity of SMP resistance other than layer 1 and 2 briefly?**

Some details have been added in the text to describe the penetration resistance heterogeneity in dry snow. We also added boxes indicating the approximate location of layers 1 and 2, so that the reader can relate more easily to Fig. 3.

*--- CHANGES IN THE MANUSCRIPT (l. 219-225) ---*

*Daily SMP measurements enable to more clearly identify the temporal and spatial variability of ice formation. Figure 6 represents the evolution of penetration resistance from 1 February to 19 April, with a scale from 0 N to 2 N to highlight variations in dry snow. The deep MFcr is visible in the middle of a low resistance depth hoar layer, at approximately 20 cm. In February and March, the highest values in the middle of the snowpack correspond to dense layers of faceted crystals (Fig. 3). In March, the buried surface hoar of layer 1 is marked by a lower penetration resistance than the surrounding faceted crystals. Layer 2 exhibits less heterogeneity with surrounding layers. Overall, the penetration resistance increases substantially from the end of March on with the progressive wetting, particularly at the top of the snowpack where many melt-freeze crusts form.*

**Line 196-199 Confirmation of melt-freeze crust on buried surface hoar seems valuable observation result. Considering the theory, a capillary barrier forms on surface hoar due to low suction of surface hoar and froze later. Is it confirmed past study or observations? If so, please add the reference.**

As highlighted by the reviewer, our observation of ice formation at a layer interface corresponding to buried surface hoar is consistent with the theory of capillary barriers. However, we have no knowledge of past studies reporting such observations.

**Fig 10-13 Please add a and b in the figure.**

Done.

**Line 247. I felt that the Fig. 12b is strange. In Fig. 10b, water ponds at 70-80cm height (on the freezing layer) on 11 February. On the other hand, despite the frozen layer was not shown in Fig 12a, water ponds at the same place. I guess that ice exists in the ice reservoir which was not shown in Fig. 12, and it affects this ponding. If my guess is correct, ice content in the ice reservoir had better be shown in some way. Also, can you describe the influence of the ice content in the ice reservoir on water percolation? In 2.2.2, formation of ice in ice reservoir is described. But influence of it on water transport should be also explained.**

Indeed, Fig. 10b (RE/PF configuration) and Fig. 12b (RE/PF/IceR configuration) show similar water ponding in the preferential flow domain. The water ponding at this layer interface is due to the fine-over-coarse grain structure. The melt-freeze crust in the RE/PF simulation is a consequence of the water ponding, with water transferred and freezing in the matrix. This ice goes into the ice reservoir in the RE/PF/IceR simulation, which explains the different grain types (Fig. 10a and 12a). The ice reservoir, however, does not affect the water transport. More generally, the microstructural changes in SNOWPACK are assessed via the ice and liquid water content in matrix domain, independently of the water in the preferential flow domain and the ice kept in the ice reservoir. This point has been clarified in the revised manuscript.

About the influence of the ice reservoir on water transport, please also see our answer to your first comment.

*--- CHANGES IN THE MANUSCRIPT (l. 279-289) ---*

*Simulations were also performed with the ice reservoir parameterization (RE/PF/IceR) to assess its ability to improve the simulation of ice layer formation compared to the previous simulations. Figure 12 shows the grain shape and liquid water content in the PF domain for the month of February. Similarly to the RE/PF simulation, the fine-over-coarse grain structure leads to water ponding in the PF domain at layer 1. But contrary to the RE/PF simulation, no melt-freeze crust forms at layer 1 (Fig. 12a): the water leaving the PF domain and refreezing is in too low quantity to be considered as representative of the mean state of the snowpack in this layer, it is thus stored in the ice reservoir. The fine-over-coarse grain transition forming a capillary barrier is preserved. Note that liquid water content in the PF domain (Fig. 12b) is almost not modified compared to the RE/PF simulation (Fig. 10b). Liquid water transport is similar, and in particular the vertical spreading of water ponding, but ice in the reservoir is concentrated at the capillary barrier. The ice kept in the reservoir has indeed no effect on water transport and microstructural changes in the matrix. At the end of February, less melt-freeze crusts are formed than in the RE/PF simulation, even though the ones surrounding layer 2 are also thicker than observed.*

**Line 325 (4 discussion): This study is attractive for researchers of water transport mechanisms using laboratory experiments and 2D or 3D models. Also, collaboration with them enhances the value of this study in terms of cover some assumptions and make physical evidence. I expect the observation data will be opened. Also, during conducting this study, if the author came up with the idea that could be done by a laboratory experiment or a 2D or 3D model, suggesting ideas in discussion will be a good information.**

As stated in the "Code and data availability" section: the dataset used in the paper will be available on the EnviDat database (doi will be provided). The ice reservoir parameterization for SNOWPACK will be available on models.slf.ch. Deposits will be ready with the final manuscript version.

Ice reservoir simulations call for other experiments on large snowpack samples, similar to Yamaguchi et al. (2018), focusing on the formation of heterogeneous ice lenses due to preferential water flow, and possibly providing further accurate validation data. These suggestions have been added in conclusion of the revised manuscript.

*--- CHANGES IN THE MANUSCRIPT (l. 399-406) ---*

*These simulations highlighted the relevance of detailed snow-cover models for the modelling of complex phenomena like deep ice layers formed by preferential water flow, since an accurate representation of the snow microstructure is necessary. Recent advances in preferential flow observations and modelling could contribute to strengthen water transport representation. This* study *also underlined the importance of comprehensive observation datasets for the validation of complex snow models. Collecting high-resolution data over more winter seasons* will further *improve the understanding of deep ice layer formation, particularly concerning their density, their impermeability and their evolution in the late melting season. Ice reservoir simulations also call for further experiments on large snowpack samples, similar to Yamaguchi et al. (2018), focusing on the formation of discontinuous ice lenses due to preferential water flow.*

---

## Author Comment (AC2) · 25 Jun 2020

**Answer to Dr. Avanzi**

We thank Dr. Avanzi for his insightful comments. Please find our detailed response to the issues raised by the reviewer below. His comments are in bold while our answers appear in blue. Changes in the manuscript appear in red (lines correspond to the marked-up manuscript). Please also refer to the marked-up manuscript.

**This study combines a valuable dataset made by snow-pit, SnowMicroPen, upGPR, lysimeter, and weather-snow measurements with improvements to the liquid-water transport scheme of SNOWPACK to study the important, but still poorly understood topic of deep ice-layer formation due to preferential flow. Results are promising, in that they show increased realism of the dual-domain scheme of SNOWPACK in reproducing ice-layer formation. Another interesting aspect of the paper is the overview of insights from the observational dataset and in particular Section 3.1.3.**

**My assessment is that the paper is interesting for readers of TC and certainly moves our understanding of ice layers forward. Yet, I do have several comments that may be useful to enhance the clarity and analyses of the manuscript (see below), so I recommend the Editor accept the manuscript after minor revisions.**

**GENERAL COMMENTS**

**1. The manuscript sometimes reads like a case study (Title, Abstract), and sometimes as an incremental contribution to Wever et al. 2016. I do believe that there are several novel points in paper, some of which are discussed at lines 49ff, while some others emerge in Section 2.2.2. Overall, however, these novel points remain somewhat implicit for general readers. I encourage authors to elaborate on framing both in the Introduction and in the Discussion to better highlight these points of novelty.**

The paper has been initially presented as a case study (starting from the title), because the comprehensive observation dataset only covered one year and one site. But the reviewer is right to underline that presenting it as a case study restricts the actual scope of the paper including several other novelties. In the revised manuscript, we endeavoured to better highlight these novelties in the title, abstract and introduction.

*--- CHANGES IN THE MANUSCRIPT (title) ---*

*Deep ice layer formation in an alpine snowpack: monitoring and modelling*

*--- CHANGES IN THE MANUSCRIPT (abstract, l. 1-14) ---*

*Ice layers may form* *deep* *in the snowpack due to preferential water flow, with impacts on the snowpack mechanichal, hydrological and thermodynamical properties. This* *detailed* *study at a high-altitude alpine site aims at monitoring their formation and evolution thanks to the combined use of a comprehensive observation dataset at daily frequency and* *state-of-the-art snow-cover modelling with improved ice formation representation. In particular, daily SnowMicroPen penetration resistance profiles enabled to better identify ice layer temporal and spatial heterogeneity when associated with traditional snowpack profiles and measurements, while upward-looking ground penetrating radar measurements enabled to detect the water front and better describe the snowpack wetting when associated with lysimeter runoff measurements.* *A new ice reservoir was implemented in the one-dimensional SNOWPACK model, which enabled to successfully represent* *the formation of some ice layers when using Richards equation and preferential flow domain parameterization,* *during winter 2017. The simulation of unobserved melt-freeze crusts was also reduced. These improved results were confirmed over 17 winters.* *Detailed snowpack simulations with snow microstructure representation, associated with high-resolution comprehensive observation dataset were shown relevant for studying and modelling such complex phenomena, despite limitations inherent to 1D modelling.*

*--- CHANGES IN THE MANUSCRIPT (introduction, l. 53-77) ---*

*The representation of ice layer formation in snow-cover models remains very challenging because it depends on an accurate description of the snow microstructure and water transport. Currently, most snow-cover models don't take into account the processes of ice layer formation. Quéno et al. (2018) recently modelled ice formation on the snowpack surface due to freezing precipitation in the detailed snow-cover model Crocus. Wever et al. (2016) included for the first time in a detailed snow-cover model the process of deep ice layer formation due to preferential water flow. Using the dual-domain implementation for water percolation in a sub-freezing snowpack, they investigated ice layer formation at an alpine site, comparing manual snow profiles recorded every two weeks over 16 winter seasons with SNOWPACK simulations. Nevertheless, many research gaps remain open about deep ice layer formation in the snowpack. In particular, the present study aims at answering the following questions:*

*– Can daily SnowMicroPen measurements improve the monitoring of ice layers in natural snowpacks over traditional snowpack profiling?*

*– How well can 1D snow-cover models represent a multi-dimensional process like ice formation due to preferential flow?*

*– Can spatially discontinuous ice lenses be parameterized in a 1D snow-cover model?*

*– Can we provide useful information on ice layer origin and evolution in alpine snowpacks for various applications, based on observations and simulations?*

*To address these research questions, we bring several novelties in a detailed study pushing forward the investigation of Wever et al. (2016). First, a comprehensive observation dataset was gathered at the same research site, in order to better determine the evolution of the snowpack and identify the formation of deep ice layers in natural conditions at a high-altitude alpine site. The originality of this dataset comes from the opportunity to monitor ice formation in natural alpine conditions during a whole winter season at daily resolution, even though the present study does not include detailed observations of preferential water flow paths. This dataset is then used for a detailed assessment of the preferential flow representation in SNOWPACK, bringing complementary insights to Wever et al. (2016) and Würzer et al. (2017). As a result, we introduce a new parameterization to improve the simulation of discontinuous deep ice formation in the SNOWPACK model.*

**2. Relatedly, novelty could also be better streamlined by explicitly identifying a few research questions to be reported at the end of the Introduction. These research questions should be broad enough to be interesting for the general public and could dramatically increase the impact of the paper.**

A new paragraph was added in the introduction to introduce the research questions that our study address and the novelties we bring to answer these questions.

*--- CHANGES IN THE MANUSCRIPT ---*

*Please refer to the answer to the first comment.*

**3. In Section 3.2.3, authors chose to restrict results in Table 1 to only simulated ice layers, without considering instances when observed ice layers were not simulated. Did I understand correctly? If so, I would suggest authors to also include these 'missed' ice layers in Table 1 to gain further insights.**

Indeed, non-simulated observed ice layers (missed events) have not been counted for the assessment overall several winter seasons. Since the validation data only consist in fortnightly visual observations, attributing several ice observations to a unique observed ice layer is difficult to establish objectively. In particular, different observers can have a different appreciation of the presence of a homogeneous ice layer. On the contrary, simulated ice layers are much less ambiguous to identify because they

persist in time. This issue highlights the high benefits of comparing simulations to our comprehensive observation dataset including daily SMP observations.

This limitation has been pointed out in the revised manuscript.

*--- CHANGES IN THE MANUSCRIPT (l. 310-313) ---*

Missed events (observed ice layers that are not simulated) are not counted. Indeed, attributing fortnightly visual ice observations to a unique observed ice layer can be very ambiguous, contrary to simulated ice layers that persist in time.

*--- CHANGES IN THE MANUSCRIPT (l. 341-343) ---*

Finally, the difficulty to identify the exact date of ice formation or to attribute isolated, fortnightly ice layer observations in traditional snowpack profiles to a unique ice layer (Sect. 3.2.3) highlights the added value of the more comprehensive observation dataset used for winter 2016/2017.

**SPECIFIC COMMENTS**

**- I would recommend authors avoid wording like 'case study' in the Title and elsewhere in the manuscript.**

Consistently with our answer to the first general comment, wording like "case study" has been removed in the revised manuscript. The title has been reworded to highlight more explicitly the scope of the study: "Deep ice layer formation in an alpine snowpack: monitoring and modelling".

**- In the abstract, I would report that the new parametrization allowed authors to significantly reduce overestimations of melt-freeze crusts (see Section 3.2). More generally, to me the abstract seems a little imbalanced toward explaining data and methods rather than findings and insights.**

The abstract has been modified, including the reviewer's suggestion.

*--- CHANGES IN THE MANUSCRIPT ---*

Please refer to the answer to the first comment.

**- Line 49: Between the previous paragraph (lines 23-48) and this one about your study (lines 49ff), I would add one more paragraph to specifically discuss previous attempts of modeling ice layers and why more research on this topic is needed. What are the knowledge gaps that you are trying to fill with this impressive field campaign and your new parametrization of ice layers? This paragraph should help formulate research questions and therefore better make the case of this paper, especially since the discussion about snow models between lines 40 and 48 is general and not really focused on ice layers.**

As suggested by the reviewer, a new paragraph was added in the introduction to highlight that many research gaps remain open in terms of ice layer representation in snowpack models. It is followed by research questions, that novelties of our study try to address.

*--- CHANGES IN THE MANUSCRIPT ---*

Please refer to the answer to the first comment.

**- Line 68: how are measurements along these three corridors managed? Do you use one corridor every other week, or simply use the same corridor every other week as long as you reach the end of it?**

Measurements were performed moving continuously along each corridor, at daily steps for the SMP measurements and weekly steps for the traditional profiles, and changing corridor once the end of the current corridor is reached. We made this detail clearer in the revised manuscript.

--- CHANGES IN THE MANUSCRIPT (l. 86-88) ---

Traditional snowpack profiles were performed during the entire season every week (or every two weeks at the beginning and the end of the season) along three corridors, moving continuously along each corridor and turning into the next one once the end of the previous one was reached.

**- Line 74: maybe cite Wever et al. 2014 when you introduce the snow lysimeter since Wever et al. 2014 also described the specific design of this instrument.**

Reference added.

**- Line 79: get rid of -> remove**

Corrected.

**- Line 100: why not initializing the simulation on a day with no snow on the ground?**

The simulations were initialized with an observed early winter snowpack in order to get the most realistic snowpack base for simulating later water percolation and ice layer formation at hydraulic barriers. Indeed, we want to focus on the assessment of the simulation of these late winter and spring processes, hence mitigate all other potential sources of uncertainties. In particular, excessive water percolation in the simulated shallow autumn snowpack may produce an unrealistic snowpack base affecting later runoff simulations.

We made it more explicit in the revised manuscript.

--- CHANGES IN THE MANUSCRIPT (l. 123-125) ---

The initialization date was chosen early enough to assess the model ability to simulate the microstructure evolution as well as water percolation, but avoiding early season modelling errors, for example, formation of unobserved basal melt-freeze layers.

**- Line 106: did you still allow for mixed rain-snow or all-rain events? If yes, using which phase-partitioning method? I assume these are rare at WFJ, but still worth discussing given that the paper focuses on liquid-water-transport schemes**

Rain events were not of concern for the period investigated at Weissfluhjoch site (2536 m.a.s.l.). A simple 1.2°C air temperature threshold was used to distinguish snow events (amount determined from measured snow depth increment) and rain events (amount determined from undercatch-corrected precipitation gauge measurements). It is now mentioned in the revised manuscript.

--- CHANGES IN THE MANUSCRIPT (l. 129-130) ---

In addition, for air temperatures above 1.2 °C, undercatch-corrected precipitation gauge measurements are considered as rain.

**- Line 119: are carried out –> were carried out**

Corrected.

**- Line 125: could you add more details about how water flowing back to the matrix domain depends on the freezing capacity of the matrix domain? As a general note about this Section, I am missing some discussion on whether the ice reservoir affects hydraulic properties of the snow layer. This is important given that ice lenses often represent a hydraulic barrier.**

We clarified the sentence about water flow from preferential domain to matrix domain. The reference to Wever et al. (2016) should allow the reader to get more details in the specified paper.

The ice reservoir has no impact on the hydraulic properties of the snow layer. Indeed, the ice reservoir mimicks the local presence of ice lenses: representing their local hydraulic barrier effect may be out of scope for our 1D simulations, given our limited knowledge and data about local ice lenses hydraulic effect. This limitation has been clarified in the section describing the ice reservoir parameterization.

 *--- CHANGES IN THE MANUSCRIPT (l. 147-176) ---*

*A new parameterization of ice layer formation due to preferential flow was implemented as a complement to the RE/PF scheme. It is summarized in Fig. 2. In the RE/PF scheme, when the saturation in the preferential flow domain exceeds the threshold $\Theta_{th}$, water flows back to the matrix domain. First, a volume of water corresponding to the available freezing capacity is instantly frozen and added uniformly to the ice content of the matrix domain. Ice lenses, to the contrary, may only form locally at the base of the flow fingers. If the threshold is still exceeded then, saturations in both domains are equalized (Wever et al., 2016).*

*To better reproduce the formation of continuous ice layers from discontinuous and growing ice lenses, we developed an ice reservoir parameterization. The water normally transferred from the preferential flow domain to the matrix domain that freezes instantly is stored in an ice reservoir (step 4 in Fig. 2), instead of being added to the ice volumetric content of the matrix. The ice reservoir is representative of the volumetric content of ice lenses (i.e. spatially discontinuous ice) in a given layer. The transferred water that does not freeze goes in the matrix domain, i.e. is spread homogeneously (step 5 in Fig. 2).*

*Furthermore, the saturation threshold in the PF domain (Wever et al., 2016) was chosen as a simple solution to the unability of Richards equation to model the saturation overshoot present in the tip of flow fingers (DiCarlo, 2007). This simple parameterization can lead to inconsistencies due to the vertical discretization of the simulated snowpack. After water has been transferred to the matrix at the layer corresponding to the finger tip (i.e. where the saturation threshold was exceeded), the highest saturation is then reached more likely at the layer above, where no water transfer occurred, because water percolation from this layer to the finger tip layer only occurs at the next time step. Because of that, the water transfer from PF domain to matrix domain may spread over too many layers, instead of being concentrated in the lowest layer (i.e. the tip of the flow finger). To overcome this issue, the ice reservoir was cumulated in the lowest layer. When the ice volumetric content of the cumulated ice reservoir added to the ice volumetric content and water volumetric content of the associated matrix layer exceeds the corresponding ice density threshold of 700 $kgm^{-3}$ in SNOWPACK, there is enough ice to consider it as horizontally homogeneous: the ice content of the cumulated ice 160 reservoir is then transferred to the associated matrix layer (step 6 in Fig. 2). As long as it is kept in the ice reservoir, the forming ice has no effect on the water transport in the matrix domain that still follows the RE/PF scheme (Wever et al., 2014, 2016). Furthermore, we neglect any impact the ice reservoir, which is interpreted as ice lenses, may have on hydraulic properties (e.g. local hydraulic barrier effect). Simulations with the ice reservoir parameterization are called RE/PF/IceR hereafter.*

*The implementation of the ice reservoir parameterization in the SNOWPACK source code is publicly available (see Code and data availability Section).*

**- Line 136-137: It is a little unclear why the highest saturation should be reached more likely at the layer above, where no water transfer occurred.**

It is actually an artefact of both the vertical layer discretization and time discretization of SNOWPACK when solving preferential water transport. Under a continuous water input in PF domain from the top of the snowpack, the saturation threshold is reached first at the layer just above the capillary barrier (layer a). The second layer above (layer b) may not reach the threshold, but has a saturation close to the one of layer a. Water transfer occurs in layer a at this time step t0. However, water transfer from layer b to layer a will only occur at the next time step t1. At the end of t0, layer b is the layer with the highest saturation. At t1, layer b also receives water input from above layers, and is at that time, the most likely layer to exceed the saturation threshold.

The time discretization artefact is now mentioned, but we tried to keep this explanation brief in the text.

*--- CHANGES IN THE MANUSCRIPT (l. 162-165) ---*

*After water has been transferred to the matrix at the layer corresponding to the finger tip (i.e. where the saturation threshold was exceeded), the highest saturation is then reached more likely at the layer above, where no water transfer occurred, because water percolation from this layer to the finger tip layer only occurs at the next time step.*

**- Line 178-179: In Katushima et al. 2013 and then Avanzi et al. 2016, it is showed that isothermal conditions may coexist with preferential flow. Given that snow pits showed vertical structural heterogeneity after April 9 (see Figure 3) and Hirashima et al. 2019 have showed that matrix flow is coupled with structurally homogeneous conditions, I would assume that matrix flow started well after April 9 at this site.**

Isothermal conditions associated with runoff are indeed not a proof of matrix flow, as shown in these papers. This part of the sentence has been removed.

**- Figure 6: maybe add the same black boxes as in Fig. 3 to denote location of ice layers? Also, the color scale is a little unclear to me: Figure 7 shows that resistance is locally well above 2N. Are all values above 2 N reported with the same color as 2 N?**

Boxes indicating the approximate location of layers 1 and 2 have been added to Fig. 6.

The purpose of Fig. 6 is to highlight the penetration resistance variations in dry and wet snow between the beginning of February and end of April. It provides a context to introduce the study of localized layers with much higher penetration resistances (Fig. 7-8-9). The high values reported in Fig. 7 are rare. Values higher than 2 N are indeed reported with the same color as 2 N.

**- Line 198: how does this manual picking selection works and what is its uncertainty?**

The SMP profiles of a given day are superimposed on the closest traditional profile and penetration resistances are associated to observed snow layers given their grain type and hand hardness index. A given SMP profile is also compared to the other SMP profiles of the same day and of the previous and next day, which enables to identify consistent patterns. In the absence of an objective layer picking technique, this method remains subjective, and its uncertainty is difficult to evaluate.

The process has been more explicitly explained in the revised manuscript.

*--- CHANGES IN THE MANUSCRIPT (l. 231-236) ---*

*The penetration resistance measured at these two layers was tracked in the SMP profiles. To identify these layers, all SMP profiles were superimposed on the traditional profile observed at the closest date. Penetration resistances were associated to observed snow layers given their grain type and hand hardness index. To identify consistent patterns, the SMP profiles of a given day were also compared among each other and with those of the previous and next day. The value of SMP penetration resistance in these layers was then visually picked. Figure 8 shows an example of this procedure for two SMP profiles of 4 April 2017.*

**- Line 205: may this also suggest that the ice layer was subjected to periodical melt and freeze?**

We don't think that melt-freeze cycles could be effective at such depth in the snowpack. It would rather correspond to a heterogeneity of the ice layer that was visually observed.

**- Line 275: Could authors elaborate on reasons why the simulated ice formation date is in average 22 days earlier than the observation interval for the RE/PF scheme and 4 days earlier 275 for the RE/PF/IceR scheme?**

The too early ice formation with the RE/PF scheme is consistent with previous findings of Wever et al. (2016) who showed that earliest season snowpack runoff from preferential flow is overestimated in simulations. Ice layers form in average 18 days later in RE/PF/IceR simulations because the ice transits through the ice reservoir before being transferred to the matrix, which delays its formation as homogeneous layer. It has been clarified in the revised manuscript.

*--- CHANGES IN THE MANUSCRIPT (l. 314-323) ---*

*Overall, the addition of the ice reservoir parameterization to the RE/PF scheme enables to form more ice layers, with a higher number of hits (15 against 6, with a one month tolerance) and a lower number of false alarms (1 against 6). The simulated ice formation date is in average 22 days earlier than the observation interval for the RE/PF scheme and 4 days earlier for the RE/PF/IceR scheme. The too early ice formation with the RE/PF scheme is consistent with the overestimation of simulated early season snowpack runoff from preferential flow as suggested by Wever et al. (2016). It is logically delayed in RE/PF/IceR simulations because the ice transits through the ice reservoir before being transferred to the matrix. The RE/PF/IceR configuration also mitigates the number of unobserved early melt-freeze crusts compared to the RE/PF configuration (not shown), as already highlighted for season 2016/2017 (Fig. 10 and Fig. 12). However, a very high number of ground ice layers were simulated (17 for the RE/PF/IceR scheme against 8 for the RE/PF scheme). This number is probably excessive, and due to possible snow-soil boundary effects.*